# Aviation soot interactions with natural cirrus clouds are unlikely to have a significant impact on global climate

Mattia Righi[1], Baptiste Testa[2,*], Christof G. Beer[1], Johannes Hendricks[1], and Zamin A. Kanji[2]

[1]Deutsches Zentrum für Luft- und Raumfahrt (DLR), Institut für Physik der Atmosphäre, Oberpfaffenhofen, Germany
[2]Institute for Atmospheric and Climate Science, ETH Zürich, Zürich, Switzerland
[*]now at: UMR TETIS, AgroParisTech, Maison de la Télédétection, Montpellier, France

**Correspondence:** Mattia Righi (mattia.righi@dlr.de)

**Abstract.** The impact of aviation soot on natural cirrus clouds is considered the most uncertain among the climate impacts of the aviation sector. In this study, a global aerosol-climate model equipped with a cirrus parametrisation is applied to quantify the impact of aviation soot on natural cirrus clouds and its resulting climate effect. For the first time, the cirrus parametrisation in the model is driven by novel laboratory measurements specifically targeting the ice nucleation ability of aviation soot, thus enabling an experimentally-constrained estimate of the aviation-soot cirrus effect. The results indicate no statistically significant impact of aviation soot on natural cirrus clouds, with an effective radiative forcing of $-6.9 \pm 29.8 \, \mathrm{mW\,m^{-2}}$ (95% confidence interval). Sensitivity simulations conducted to investigate the role of other ice nucleating particles (INPs) competing with aviation soot for ice supersaturation in the cirrus regime (soot from sources other than aviation, mineral dust and ammonium sulphate) further show that the impact of aviation soot remains statistically insignificant also when the impact of these other INPs on cirrus is reduced in the model. Acknowledging that the complexity of the soot cirrus interaction is associated with uncertainties, the model results supported by dedicated laboratory measurements suggest that the climate impact due to the aviation soot cirrus effect is likely negligible with no statistical significance.

## 1 Introduction

The impact of aviation-emitted soot particles on natural cirrus clouds is highly uncertain and best estimates on the resulting climate effect are not available to date (Lee et al., 2021, 2023). This is due to the complex and poorly constrained physical processes involved in the interactions between aviation soot and cirrus clouds and due to the inherent challenges in representing these processes in global climate models. The formation of ice crystals in the cirrus regime ($T \lesssim 235$ K) can occur either homogeneously from solution droplets or heterogeneously, i.e. in the presence of INPs, such as mineral dust or soot (Vali et al., 2015). The homogeneous freezing of liquid solution droplets takes place when the ice supersaturation is sufficiently high ($S_i \gtrsim 1.4$; Koop et al., 2000) and usually results in the formation of a relatively large number of small ice crystals, while heterogeneous freezing can occur at lower supersaturations ($S_i \gtrsim 1.1$) forming fewer and larger ice crystals, due to the scarcity of INPs in the upper troposphere compared to liquid solution droplets (DeMott et al., 2010). These two ice nucleation processes compete with each other for ice supersaturated water vapour. The microphysical (ice crystal size and number) and radiative

properties of the resulting cirrus are controlled by this competition. Different INP types also compete with each other for the heterogeneous formation of ice crystals, also influencing cirrus properties (DeMott et al., 2003; Kärcher and Lohmann, 2003).

Considering the impact of aviation soot as an INP further complicates this picture and makes it challenging for global models to robustly quantify its impact. For this task, the models need to be able to represent the distribution and properties not only of aviation soot, but also of other INP types, such as mineral dust, soot from sources other than aviation (hereafter, background soot) and, as shown by recent studies (Beer et al., 2022), ammonium sulphate and, possibly, organic aerosols. Moreover, models need to be equipped with parametrisations for cirrus clouds accounting for homogeneous and heterogeneous ice formation and their competition, with a realistic representation of vertical updrafts controlling the cooling rates and the supersaturation. The representation of such updrafts is particularly challenging for cirrus, as it mostly occurs at spatial scales which cannot be resolved by global models (Lohmann and Kärcher, 2002) limited by their coarse spatial resolution of the order of 100 km. Orographic gravity waves are also relevant in this regard, as they can contribute substantially to cirrus coverage, promoting homogeneous freezing due to their strong updrafts (e.g. Barahona et al., 2017). An additional complication comes from the need to isolate the impact of aviation soot on cirrus clouds from that of other INPs and to distinguish it from the internal model variability, which poses additional statistical challenges.

This partly explains why only few modelling studies have so far attempted to quantify this effect and why no consensus has been reached on the resulting effective radiative forcing (ERF). Several studies based on different versions of the NCAR CAM model (Liu et al., 2009; Penner et al., 2009; Zhou and Penner, 2014; Penner et al., 2018; Zhu and Penner, 2020) reported large ERF from the aviation soot-cirrus effect, in the range of $-350$ to $+260\,\mathrm{mW\,m^{-2}}$, depending on the model version and on the assumption of the ability of aviation soot to nucleate ice in the cirrus regime. These large estimates are not supported by any of the other studies on this effect: Hendricks et al. (2011), with the ECHAM4 model, Gettelman and Chen (2013), with the CAM5 model, and McGraw et al. (2020), with the CESM2 model, all reported a statistically non-significant effect. Righi et al. (2021) quantified the aviation soot-cirrus effect with the EMAC model for a range of assumptions on the ice nucleation ability of aviation soot, also finding statistically non-significant results in most cases and a small ERF of $-35$ to $-23\,\mathrm{mW\,m^{-2}}$ when assuming a strong ice nucleation ability of aviation soot. Using a cirrus column model at high resolution, Kärcher et al. (2021) found no fundamental difference in the optical depth of soot-perturbed and homogeneously-formed cirrus, concluding that global models may have overestimated the aviation-soot cirrus effect. In a follow-up study, Kärcher et al. (2023) showed that the ice nucleation of aviation soot is prevented by mineral dust INPs at typical atmospheric conditions. However, Urbanek et al. (2018) used lidar measurements to report higher particle linear depolarization ratios for cirrus clouds along flight corridors over Europe, arguing that this could be traced back to heterogeneous freezing on aviation soot particles. A similar hypothesis was proposed by Li and Groß (2021), Li and Groß (2022) and Groß et al. (2023) to explain the observed changes in frequency and properties of cirrus clouds in areas affected by air traffic.

The ice nucleation ability of aviation soot assumed in the above model studies were derived from aviation soot surrogates or from theory. Yet, the ice nucleation ability of soot particles has proven to be very sensitive to the source of emission (Mahrt et al., 2018; Bhandari et al., 2019; Brooks et al., 2014; Möhler et al., 2005; Koehler et al., 2009; Gao et al., 2022b; DeMott

et al., 1999). The properties of soot particles derived from surrogates could therefore be considerably different from those emitted by aircraft engines.

The present study was motivated by recent measurements of the ice nucleation ability of aircraft soot particles by Testa et al. (2024a, b), where ground-based sampling of soot particles from modern in-use commercial aircraft engines were conducted, followed by in-line ice nucleation measurements of the sampled aviation soot. To the best of our knowledge, these constitute the most representative measurements on the ice nucleation ability of aviation engine emitted soot. The overarching results of the studies by Testa et al. is that aviation soot requires saturation levels close to those for homogeneous ice nucleation of solution

droplets, making it a poor INP for cirrus formation. Here, for the first time, we use the results of these measurements to drive numerical simulations with a state-of-the-art global aerosol-climate model, thus providing the first experimentally-constrained quantification of the aviation-soot cirrus effect. We show that the effect is very small, exhibiting no statistical significance at the 95% confidence level. Sensitivity simulations reducing the effectiveness of other INPs competing with aviation soot for heterogeneous freezing in the cirrus regime also do not allow to isolate a significant effect, even under the strong assumption

to enhance the role of aviation soot at the expense of other INPs in the ice formation process. Therefore, we conclude that it is unlikely that the aviation-soot cirrus effect plays a significant role in the context of the climate impact of aerosol-cloud interactions.

## 2    Methods

### 2.1    Measurements

The ice nucleation ability of aircraft engine soot was determined experimentally as detailed in Testa et al. (2024a, b). Briefly, the soot particles were sampled from commercial aircraft engines at the aircraft engine maintenance facility, SR Technics, at Zürich airport. Particles from Pratt and Whitney (P&W) and CFM International engines, together representing more than 70% of the global fleet, were examined (Testa et al., 2024a). The measurements were performed on emissions from five different engine models that were all fueled with standard jet fuel (Jet A-1). Polydisperse aircraft soot particles were sampled with

mode-diameter (number concentration) ranging from 80-450 nm, and the engines running at various thrust levels, including cruise thrust. Although slightly larger than what was measured at flight altitude (Moore et al., 2017), the physicochemical properties of the sampled aircraft soot particles are believed to be representative of in situ aircraft soot (see also the discussion in Sect. 4). Contrail-processing of the sampled aircraft soot particles was simulated with custom-designed ice nucleation chambers (see, e.g., Mahrt et al., 2020; Testa et al., 2024b; Gao et al., 2022b) and their ice nucleation ability was subsequently

quantified. We note that our setup does not mimic the dynamical processes and rates relevant for contrail formation exactly. For example, exhaust temperature dropping rapidly ($< 1$ s) from thousands of degrees to -60 °C and pressure from approximately $10^6$ Pa to below $10^5$ Pa (Kärcher, 2018). Instead, the time scales are longer than what it would be in the atmosphere. In our set up the pressure drops from approximately $10^6$ Pa in the engine, to approximately $10^5$ Pa (atmospheric pressure in our measurement set up). The temperature also drops in three steps, from the engine temperature to the heated line 160 °C and

then to room temperature followed by a third drop from room temperature to the cloud chamber temperature. So, we do have

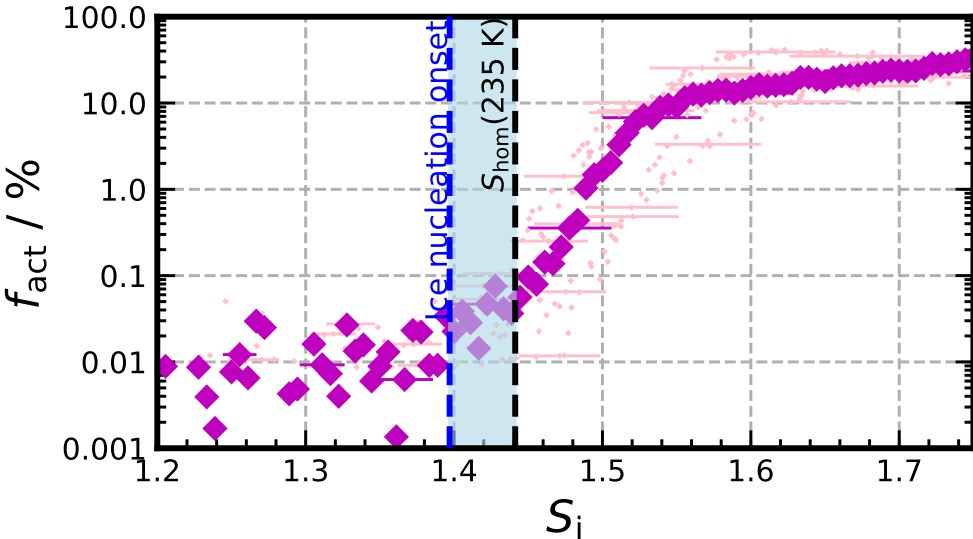

**Figure 1.** Ice nucleating active fraction (AF) of polydisperse contrail processed and bare (coating-free) soot (CS-CP-soot in Testa et al 2024b) from Testa et al 2024b as a function of $S_i$. Individual measurements are shown with the pink markers and the median AF curve is shown with the purple markers. The pink horizontal error bars represent releative humidity (RH) uncertainties on the measurements. The blue and black dashed lines correspond to the mean aircraft soot ice nucleation onset and to $S_{hom}$ at 235 K, respectively. The part of the AF curve outside of the blue shaded area cannot be considered in the model. See text for details.

similar temperature drops as would occur in the atmosphere, but they occur over a slightly longer time scale. The results of Testa et al. (2024a, b) showed that the aircraft soot is a poor INP, nucleating ice at or close to the water vapor saturation required for the homogeneous nucleation of the ice ($S_{hom}$). Only when the contrail-processed aviation soot particles are free of any coating (mainly sulphuric acid and organics), do they exhibit ice nucleation activity below that required for homogeneous ice nucleation of solution droplets. In the model we therefore apply these ice nucleation results to the insoluble soot mode of the aerosol microphysical scheme. Fig. 1 shows the active fraction (AF) curves derived from the measurement for different engine models. The blue shaded area in Fig. 1 shows the ice saturation range where aircraft soot particles can form ice crystals and compete with aqueous solution droplets and other INPs for the available supersaturated water vapor. This effective $S_i$ range is bound by $S_{hom}$ and by the ice nucleation onset of the soot particles. The ice nucleation onset was reported when the particles activated fraction exceeds the cloud chamber background noise levels ($f_{act}$=0.01%). This was derived from the median of all AF curves and estimated at $S_i = 1.397$ (Fig. 1). Lower values of $S_i$ at $f_{act}$=0.01% appear in Fig. 1 but are below the detection limit of the instrument and thus have a low confidence. For the model simulations performed in this study, the ice nucleation ability of aviation soot was considered at its onset of activation (termed critical saturation ratio, $S_{crit}$).

## 2.2 Model and model simulations

We use the EMAC global chemistry-climate model (Jöckel et al., 2010), equipped with the aerosol microphysical scheme MADE3 (Kaiser et al., 2019) coupled to a two-moment cloud microphysical scheme including a parametrisation for aerosol-induced ice formation in the cirrus regime (Kärcher et al., 2006; Kuebbeler et al., 2014). This model configuration has been extensively documented and evaluated in Righi et al. (2020) and successfully applied in several studies (Beer et al., 2022; Righi et al., 2023; Beer et al., 2024), including the assessment by Righi et al. (2021) on the aviation-soot cirrus effect under different assumptions for the ice nucleation ability of aviation soot. Soot aviation emissions are based on the CMIP6 inventory for the year 2014 (Hoesly et al., 2018), resulting in a global emission of about 10 Gg a$^{-1}$. Soot particles are assigned to the black carbon (BC) tracers of the aerosol submodel MADE3 in EMAC. Soot particle number emissions are calculated from the mass emission fluxes using the lognormal size distribution parameters by Petzold et al. (1999), obtained from in situ measurements behind aircraft at cruise altitude. Their applicability in the large-scale models is supported by the results of Mahnke et al. (2024) using the IAGOS-CARIBIC Flying Laboratory data (Brenninkmeijer et al., 2007). These lognormal parameters are also used to calculate the mass fractions of emitted soot between the Aitken and accumulation modes of the MADE3 submodel in EMAC. Detailed descriptions about the EMAC model configuration adopeted in this work can be found in Righi et al. (2021). Since MADE3 explicitly simulates three different aerosol mixing states (soluble, insoluble and mixed; Kaiser et al., 2019), we introduced an important update in this study, by only allowing aviation soot in the insoluble modes of MADE3 to act as INP, consistent with the measurement results of coating-free soot described above indicating that only sulphur-free (uncoated) soot particles nucleate ice below the homogeneous freezing threshold, while in Righi et al. (2021) both insoluble and mixed aviation soot were allowed as INPs.

The simulations performed in this study are summarized in Table 1. The properties of aviation soot and other INPs are parametrised in the model by means of two parameters: the critical saturation ratio with respect to ice $S_{\mathrm{crit}}$ at which the INP nucleates ice and the active fraction $f_{\mathrm{act}}$ of the INP population which forms ice crystals. In all model experiments, the ice nucleation properties of aviation soot are based on the parameters measured in the laboratory experiments described in Sect. 2.1. In the reference (REF) simulation, the parameters accounting for the heterogeneous ice formation of the other INPs are the same as in Righi et al. (2021). In the NOBGSOOT simulation, the impact of background soot (i.e., soot from emission sources other than aviation) is switched off. In the NOBGSOOT+DUST5 and NOBGSOOT+DUST10 the contribution of mineral dust INPs to the immersion and deposition mode is reduced, by scaling $f_{\mathrm{act}}$ down by a factor 5 and 10, respectively. These two simulations aim to account for a potential positive bias of EMAC in the representation of mineral dust concentration in the upper troposphere (see Fig. S1 in Beer et al., 2024). Reducing the active fraction of dust INPs is a way to implicitly correct for this bias. The impact of ammonium sulphate INPs is assessed in the NOBGSOOT+AMSU simulation, which considers ice nucleation by dust INPs and ammonium sulphate INPs. The model version for this sensitivity experiment is based on the setup described in Beer et al. (2022, 2024). The ice nucleating properties of crystalline ammonium sulphate are chosen according to Ladino et al. (2014) and Bertozzi et al. (2024).

**Table 1.** Model simulations with different sets of ice nucleation parameters for the INPs competing for available supersaturation in the cirrus parametrisation of the model: aviation soot and background soot (backgr. soot) in the deposition mode, mineral dust in the immersion mode, mineral dust in the deposition mode, and ammonium sulphate (amm. sulph.) in the deposition mode. Background soot properties are taken from Hendricks et al. (2011). M06 refers to the temperature-dependent parametrisation for mineral dust in the deposition mode by Möhler et al. (2006). RIGHI21 represents the S14F01 simulations conducted by Righi et al. (2021), which assumed particularly low ice nucleation ability for aviation soot. For each simulation, a corresponding baseline simulation neglecting the impact of aviation soot on cirrus clouds is performed (i.e., setting $f_{\mathrm{act}}$ of aviation soot to zero).

| Simulation | aviation soot (deposition) | | backgr. soot (deposition) | | dust (immersion) | | dust (deposition) | | amm. sulph. (deposition) | |
|---|---|---|---|---|---|---|---|---|---|---|
| | $S_{\mathrm{crit}}$ | $f_{\mathrm{act}}$ [%] | $S_{\mathrm{crit}}$ | $f_{\mathrm{act}}$ [%] | $S_{\mathrm{crit}}$ | $f_{\mathrm{act}}$ [%] | $S_{\mathrm{crit}}$ | $f_{\mathrm{act}}$ [%] | $S_{\mathrm{crit}}$ | $f_{\mathrm{act}}$ [%] |
| REF | 1.397 | 0.01 | 1.4 | 0.25 | 1.3 | 5 | M06 | | – | – |
| NOBGSOOT | 1.397 | 0.01 | – | – | 1.3 | 5 | M06 | | – | – |
| NOBGSOOT+DUST5 | 1.397 | 0.01 | – | – | 1.3 | 1 | M06 | M06/5 | – | – |
| NOBGSOOT+DUST10 | 1.397 | 0.01 | – | – | 1.3 | 0.5 | M06 | M06/10 | – | – |
| NOBGSOOT+AMSU | 1.397 | 0.01 | – | – | 1.3 | 5 | M06 | | 1.25 | 0.1 |
| RIGHI21 | 1.4 | 0.1 | 1.4 | 0.25 | 1.3 | 5 | M06 | | – | – |

All simulations cover a period of 15 years (2001–2015) preceded by a spin-up year (2000) not considered for the analysis. To assess the impact of aviation soot on natural cirrus clouds model simulations are performed pairwise, comparing each simulation with a corresponding baseline where the impact of aviation soot in the cirrus parametrisation is switched off (i.e., $f_{\mathrm{act}} = 0$). The difference of the top-of-the-atmosphere radiative fluxes between the two simulations provides then a quantification of the aviation-soot ERF. Note that this is an effective RF (and not an instantaneous RF) since it includes the effect of cloud adjustments to the aviation soot perturbation. To validate the statistical significance of the results, a paired-sample t test is applied. The results are considered significant if the null hypothesis that the paired simulations are identical can be rejected at a confidence level larger than 95%. The same methodology is applied when other model variables are evaluated, such as the aviation-soot-induced changes in ice crystal number concentration (ICNC), total water (water vapour plus ice water) and cloud frequency. To reduce the impact of the internal model variability when comparing two simulations, the model meteorology (temperature, winds, and logarithm of the surface pressure) is nudged towards the ERA-Interim reanalysis data (Dee et al., 2011) of the European Centre for Medium-Range Weather Forecasts (ECMWF) for the simulation period.

## 3  Results

No statistical significant impact of aviation soot on cirrus can be quantified for the ice nucleation ability measured in the laboratory studies (discussed in Sect. 2.1). As shown in Fig. 2, in the REF simulation, the aviation soot-cirrus effect is centered around $-6.9$ mW m$^{-2}$ but with a large 95% confidence interval ($\pm 29.8$ mW $^{-2}$) which makes this result statistically indistin-

guishable from zero. This very small effect results from the combination of a negative shortwave ERF and a positive longwave ERF of a similar magnitude (Fig. 3a,b): this is consistent with the increase in ICNC seen in Fig. 3f, possibly reducing their size

155 and hence sedimentation, resulting in a higher cirrus cloud reflectivity (i.e., more negative shortwave ERF), and in a higher water content and cloud frequency (Fig. 3g,h), both increasing the longwave ERF. Note, however, that this interpretation is hampered by the low statistical significance of the results. The impact on the homogeneous freezing fraction is negligible (Fig. 3e; see Righi et al. (2021) for the definition of this quantity), indicating that, on the global mean, aviation soot does not prevent the homogeneous formation of ice crystals, but just competes against other INPs for heterogeneous freezing. The very

160 low statistical significance of this effect, however, suggests that with such low nucleation ability, aviation soot has little chance to compete against other more effective INPs for available supersaturated water vapour: as shown in Fig. 4, only 0.04% of heterogeneously formed ice crystals stem from aviation soot, while mineral dust and soot from background sources largely dominate the heterogeneously formed ICNC at cirrus altitudes ($\lesssim$400 hPa). This result is in line with the simulation S14F01 of Righi et al. (2021), see the white bar in Fig. 2, although that assumed a factor 10 higher $f_{\text{act}}$ for aviation soot.

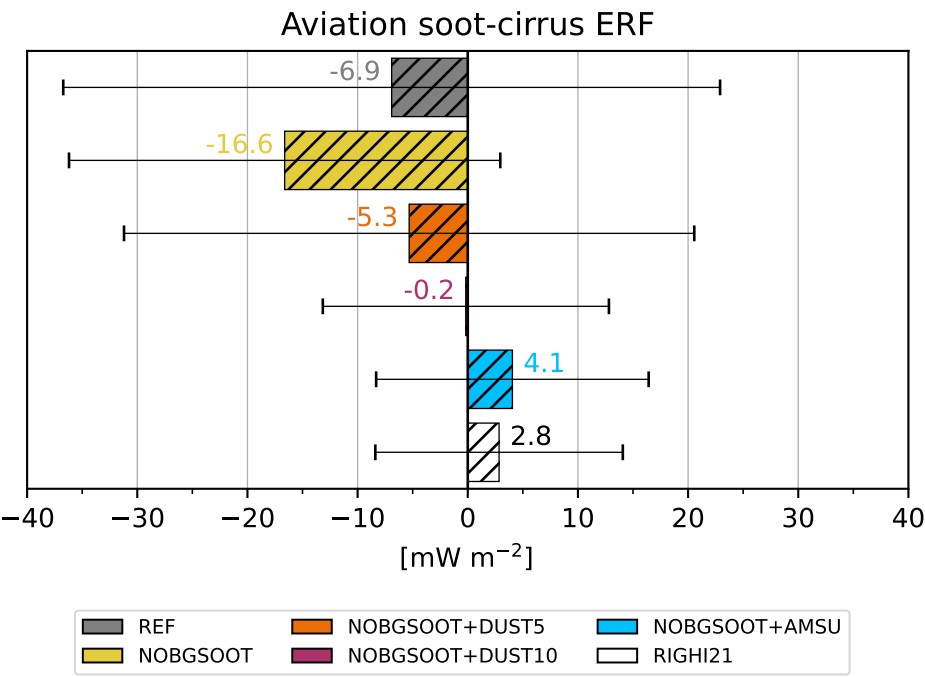

**Figure 2.** (ERF from the aviation-soot cirrus effect for the model simulations performed in this study. The error bars represent the 95% confidence interval. The average value is also shown besides each bar, in units of mW m$^{-2}$. Hatched bars indicate a statistically non-significant result, i.e. when the confidence interval crosses the zero line.

165 Given the ice nucleation properties of aviation soot are now constrained by measurements, the uncertainty on the role of other INPs on the aviation-soot cirrus effect can be assessed by varying their representation in the model simulation. This

is realized by first focusing on soot from natural and non-aviation anthropogenic sources, e.g., combustion of fossil fuels in stationary and other mobile sources, or biomass burning. As these are ground-based sources, it is reasonable to assume that the soot particles emitted by these sources are relatively aged when reaching the upper troposphere (Bond et al., 2013), i.e. the main region of interest for investigating aviation effects. As shown by several studies (see Kanji et al., 2017, and references therein) and by the measurements used in the current study, aged soot particles are very ineffective INPs and they do not play any role (or only a marginal one) in the ice formation process. Hence, they would not compete for ice formation with aviation soot in the cirrus formation process and their role may have been overestimated in Righi et al. (2021). Neglecting the impact of background soot (NOBGSOOT simulation), the effect of aviation soot indeed increases to $-16.6 \, \mathrm{mW \, m^{-2}}$ and the confidence interval is reduced ($\pm 19.6 \, \mathrm{mW^{-2}}$), but the result can still not be deemed as significant at the 95% confidence level. In the NOBGSOOT case, the shortwave and longwave ERF have opposite signs similar to the REF case, where the shortwave has a similar magnitude, but the longwave ERF is substantially reduced, resulting in a more negative ERF in the NOBGSOOT than in the REF simulation. This appears to be related to the aviation-soot-induced reduction in homogeneous freezing fraction (Fig. 3e), which may counteract the increase in ICNC seen in the REF case and result in no overall changes to ICNC. This leads to a limited impact on both cloud water content and cloud frequency and lifetime, thus contributing to the small longwave ERF. The impact of aviation soot on the heterogeneous ice formation remains, however, limited: as shown in Fig. 4b, when removing background soot from the system, its role in the process is effectively overtaken by mineral dust, mostly in the deposition mode, and the share of aviation soot remains at 0.04% as in the REF case.

Although the mineral dust ice nucleation abilities are relatively well constrained (Möhler et al., 2006; Ullrich et al., 2017), the EMAC model shows a positive bias of a factor of about 5 to 10 in the simulated concentration of mineral dust in the upper-tropospheric northern mid-latitudes (Beer et al., 2022), i.e. the region of interest for the aviation-soot cirrus effect investigated here. To implicitly correct for this bias, two additional simulations are performed reducing the active fraction of the mineral dust INPs by a factor 5 and 10 (simulations NOBGSOOT+DUST5 and NOBGSOOT+DUST10, respectively). In the first simulation, the longwave ERF increases, resulting in an aviation-soot ERF similar to the REF case: $-5.3 \, \mathrm{mW \, m^{-2}}$, again with a large confidence interval around the central value, making the effect statistically indistinguishable from zero. ICNC is also increased in this simulation, despite a reduction in the homogeneous freezing fraction (Fig. 3e,f): this could be related to the factor 5 increase in the share of heterogeneously formed INPs from aviation soot (Fig. 4c), which becomes more effective as the competition from dust INPs is reduced. As in the REF case, the increase in ICNC has an impact on both, the total water and cloud frequency (Fig. 3g,h). One would expect that further reducing the active fraction of mineral dust INPs by a factor of 10 with respect to the REF case (NOBGSOOT+DUST10) could enhance this trend, but the results actually show that the soot-cirrus ERF is reduced to almost zero in this simulation ($-0.2 \, \mathrm{mW \, m^{-2}}$), with negligible aviation-soot-induced changes in all relevant quantities (Fig. 3). A possible reason for this could be a stronger sedimentation due to fewer and much larger ice crystals, reducing ICNC (Fig. 4f), also with a smaller impact on the homogeneous freezing fraction compared to the NOBGSOOT+DUST5 case (Fig. 4e). The share of aviation-soot-induced INPs in the heterogeneously formed ICNC increased by a factor of 2 compared to the NOBGSOOT+DUST5, but mineral dust remains the dominant INP in this process (Fig. 4d).

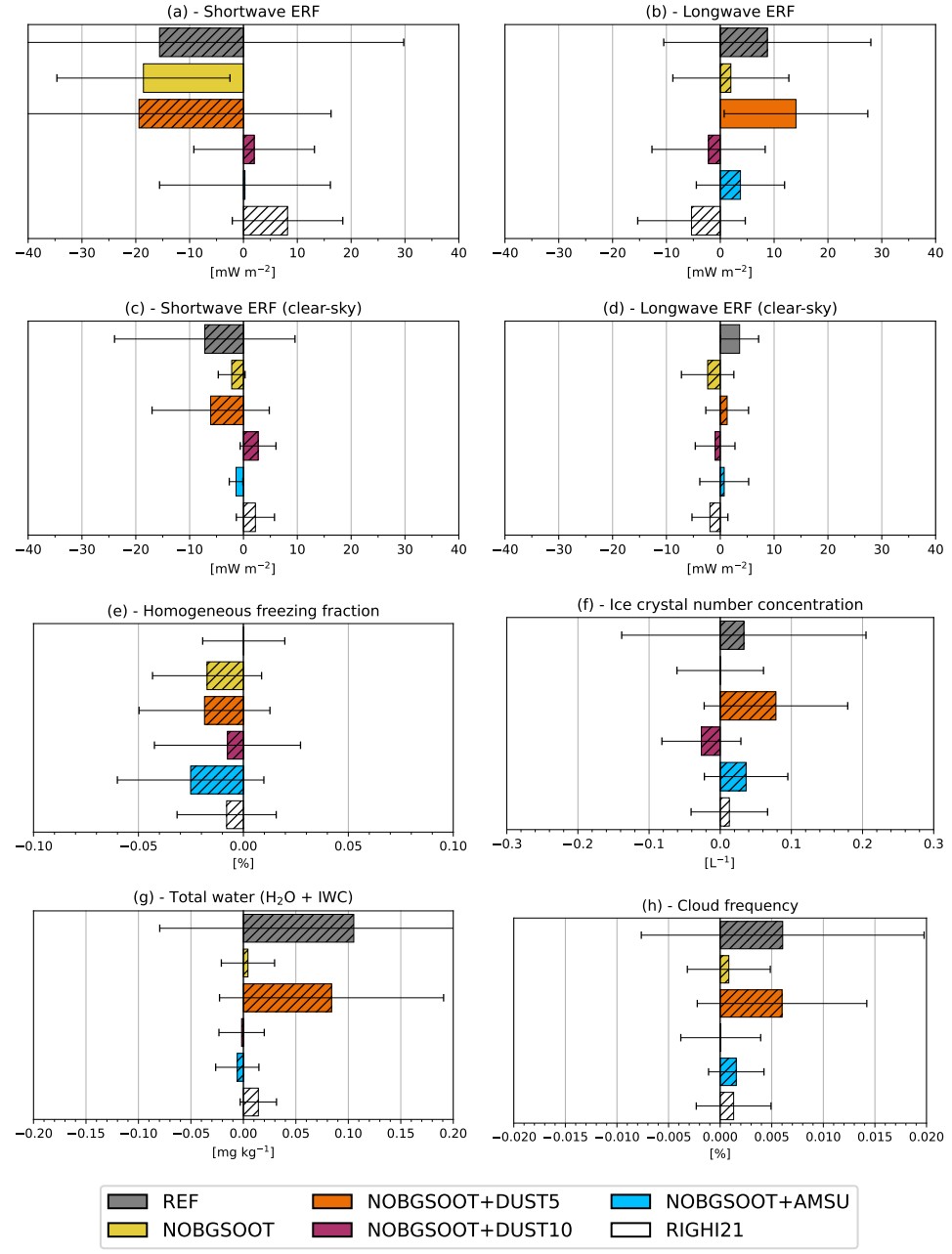

**Figure 3.** Aviation-soot-induced changes in key radiation and cloud variables. Radiative forcings are calculated at the top of the atmosphere. Other quantities are spatially averaged at cirrus altitudes (above ∼400 hPa) and over cloudy and cloud-free model grid-boxes. Percent units in the cloud frequency refer to the absolute change in frequency.

Finally, we analysed the impact of crystalline ammonium sulphate, which has been reported as an effective INP by several studies (Abbatt et al., 2006; Wise et al., 2009; Baustian et al., 2010). For the simulation NOBGSOOT+AMSU, we use the model configuration by Beer et al. (2024), which is based on the one adopted here (Sect. 2.2), with a few extensions to account for the formation process of ammonium sulphate and its ice nucleation ability in the cirrus regime. We note that the results of the NOBGSOOT+AMSU simulations are not perfectly comparable with the other simulations, since they are based on a different model configuration. However, they provide useful insights into the role of ammonium sulphate INP in the context of the aviation-soot cirrus effect. Introducing ammonium sulphate as a further INP in the system while still neglecting background soot, the aviation-soot cirrus effect is slightly positive (4.1 mW m$^{-2}$), but again not statistically significant at the 95% confidence interval. No impact of aviation soot on the shortwave ERF is found in this simulation, while the longwave ERF is slightly positive (Fig. 3a,b). As a very effective INP, ammonium sulphate effectively competes against the other ones for heterogeneous freezing, also due to its relatively large concentrations in the upper troposphere (see Fig. 5 in Beer et al., 2022). This leads to a reduction in the share of both mineral dust and aviation soot (Fig. 4e), possibly also explaining the reduction in the homogeneous freezing fraction (Fig. 3e), which is the largest across all simulations, and the increase in ICNC (Fig. 3f). Note, however, that crystalline ammonium sulphate INPs are not omnipresent, but only form after efflorescence, vanishing again after deliquescence. They are therefore present in large number concentrations and dominate the process over short periods of time, while on long temporal scales their effect is smaller.

In summary, the quantification of the aviation-soot cirrus effect with the support of novel laboratory measurements on the ice nucleating properties of aviation soot result in a non-statistically significant ERF effect for all investigated cases. The interpretation of the model results in relation to key cloud and radiation variables is substantially hampered by the very low statistical significance of almost all discussed quantities and in all simulations. It is therefore very challenging to draw a coherent picture, as the effect of aviation soot on natural cirrus clouds is very small compared to the internal model variability. This is also the case when repeating the analysis with a geographically-resolved simulation output: as shown in the maps and zonal profiles of the ERF effect provided in the Supplement, these are characterized by an extremely noisy pattern and no coherent picture emerges that could point to a significant ERF at the local level (e.g., over the North Atlantic or in other highly travelled regions of the world). The model results do not rule out the possibility of a localized effect, but they do indicate that it is likely to be small compared to natural climate variability, and therefore, the overall climate impact is expected to be minimal.

## 4 Discussion

Several previous studies based on in situ measurements and satellite observations pointed to a possible impact of aviation soot on natural cirrus clouds. Urbanek et al. (2018) performed airborne lidar measurments of cirrus optical properties over Europe and reported that half of the clouds showed high depolarization ratios, which are associated with complex ice crystal shapes and, in turn, with increased availability of INPs during the formation process. Their analysis excluded the impact of embedded contrails, but did not exclude aviation as a possible source of increased INPs. Taking advantage of the low air traffic conditions over Europe during the COVID-19 pandemic in 2020, Li and Groß (2021) used satellite lidar measurements to investigate cir-

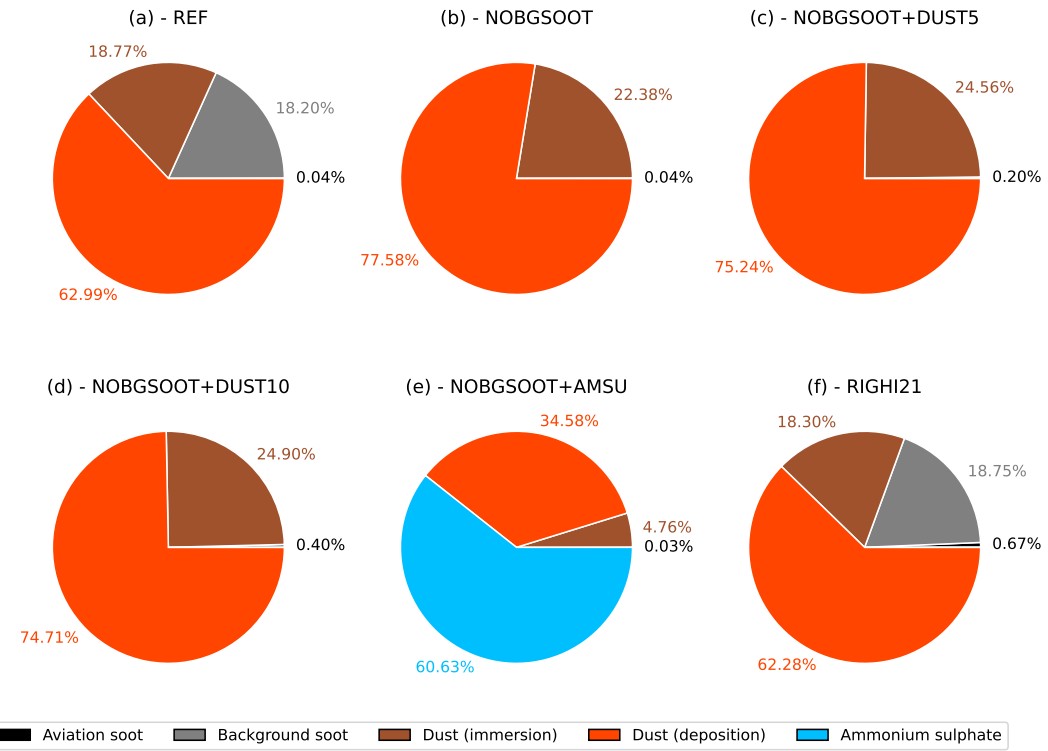

**Figure 4.** Relative share of the ice crystal number at cirrus altitudes (above ∼400 hPa) heterogeneously formed by different INP types in the simulations performed in this study. The shares are calculated considering the ice crystal number concentration calculated by the cirrus parametrisation before merging the ice formation modes in a single population and applying the ice crystals growth and sedimentation processes. They are therefore not fully representative of the ice crystal population, but provide a broad indication of the prevailing INP type in the heterogeneous ice formation process.

rus properties during this period and reported a significant reduction in depolarization ratios compared with the same months in previous years, thus supporting the hypothesis of aviation leading to increased heterogeneous freezing in cirrus clouds. In a similar analysis on the 10 years before the COVID-19 pandemic, Li and Groß (2022) found a statistically significant increasing trend in the depolarization ratio of cirrus clouds, again pointing to aviation as a source of additional INPs. Further support to this hypothesis came from the combined airborne lidar and in situ ice cloud measurements by Groß et al. (2023), comparing regions affected by aviation aerosol with unaffected regions. They found higher depolarization ratios coinciding with larger ice crystal size and lower concentrations in aviation-affected regions and proposed heterogeneous freezing on aviation soot as a possible explanation for the observed cirrus properties. Although our results seem to contradict the conclusions of the above studies, we note that our model simulations do not actually exclude an impact of aviation soot at the local level under certain conditions (e.g., specific temperature ranges), but show that this impact is not statistically significant on climatically relevant spatial and temporal scales, resulting in a likely small global climate effect. This, moreover, is consistent with the vast

majority of previous modelling studies (Hendricks et al., 2011; Gettelman and Chen, 2013; McGraw et al., 2020; Kärcher et al., 2021, 2023). We also remark that the results of the above measurements point to aviation as a possible reason for the observed changes in the optical properties of cirrus, but not specifically to soot particles. Other INP types, such as for example metallic particles released by the aircraft, could also be responsible for the observed changes in cirrus properties: Cziczo et al. (2013), for instance, found metallic particles in the residuals of ice crystals in heterogeneous-freezing-dominated cirrus, although they could not clearly separate metallic residuals from mineral dust residuals. Targeted model experiments that more closely match the conditions of the measurement campaigns could help disentangle the causes of the observed correlation, although global climate models struggle to reproduce such specific conditions at fine temporal and spatial scales. This is nevertheless worth to be investigated in follow-up studies.

The laboratory measurements used to drive the simulations in this study provide direct evidence on the poor ice nucleation ability of aviation soot. Although these measurements have been performed in laboratory conditions at ground level, they are the first measurements sampling real engine aviation soot and can be considered fairly representative of actual atmospheric conditions. As explained in Testa et al. (2024a, b), the number and size of cavities within the soot aggregates are the primary controlling factors of soot ice nucleation via pore condensation freezing. This is primarily controlled by the primary particle morphology and the aggregates size. Gas phase chemistry and particle oxidation are thought to considerably slow down as the exhaust exits the combustor chamber, due to lower temperatures in the exhaust nozzle and downstream of the engine (Wong et al., 2008). As such, primary particle properties are fixed in the combustor (Dakhel et al., 2007). Such a drop in temperature is also present in the sampling system in our laboratory experiments (thousand degrees to 160 °C), thus the primary particle overlap, size, crystallinity and oxidation are thought to be comparable to in situ emitted aviation soot particles. Besides, we note that (relatively high) temperature and pressure in the combustor ($\sim$2000 K and $\sim 10^6$ Pa, respectively; Dakhel et al., 2007; Starik, 2007) are largely driven by the engine design and thrust, hence uncorrelated to the ambient conditions. Regarding the atmospheric relevance of the soot aggregate sizes in the measurement studies (mode diameter of 80–450 nm), airborne in situ measurement of soot sizes in young aircraft plumes are scarce: Petzold et al. (1999), Petzold et al. (1998) and Twohy and Gandrud (1998) observed interstitial and contrail residual soot aggregates of 0.15–1 $\mu$m. Those large aggregates could result from the coagulation of soot aggregates trapped in the wingtip vortices (Miake-Lye et al., 1993) due to the higher soot emission index for older engine models (Lee et al., 2010; Masiol and Harrison, 2014). Coagulation of contrail ice crystals and merging of embedded soot aggregates upon sublimation of the ice crystals could also lead to larger soot aggregates. Yet, for current aircraft engines with lower emission indices, coagulation of the soot aggregate is reduced or inhibited, as shown by Moore et al. (2017), who did not observe soot coagulation, and whose measurements are similar to what would be measured at ground in engine maintenance and testing facilities (e.g., Durdina et al., 2021). Thus, under ambient conditions, coagulation is likely not expected. Hence atmospheric aircraft soot has very likely smaller mode diameters than soot aggregates sampled in the measurement study, which would further limit the potential to act as INP at RH below homogeneous freezing (size dependency of soot particles has been shown in numerous studies before, e.g. Mahrt et al., 2018; Zhang et al., 2020; Gao et al., 2022a).

We expect in situ particles to be coated with sulphuric acid and organics. To which extent the coating of the aviation soot sample in the measurement study is different from in situ aviation soot cannot be quantified, but any condensation of organics

or sulfate in the atmosphere would first condense into soot pores which inhibits ice nucleation (Gao et al., 2022c; Gao and Kanji, 2022). Thus, the ice nucleation abilities of coating-free and coated soot were quantified in the measurement study to bound the possible effect of different aviation soot mixing states. We note that a larger or smaller soot coating compared to the investigated coated soot sample would favour homogeneous nucleation, or be bound by a coating-free soot sample, respectively. Finally, as mentioned above, ice nucleation is controlled by the limited availability and morphology of inter-aggregate cavities, primarily governed by the primary particle morphology, which is not affected by the experimental setup and sampling method but rather fixed in the engine combustor.

## 5 Conclusions

Novel laboratory measurements of the ice nucleation ability of aviation soot at cirrus temperatures are used to drive simulations with a global aerosol-climate model to quantify the effect of aviation soot on natural cirrus clouds. With these measurements, the uncertainties in the ice nucleating abilities of aviation soot explored in the former assessment by Righi et al. (2021) with the same model are constrained and, for the first time, an experimentally-informed aviation-soot cirrus effect is quantified.

The model results show that the ERF effect of aviation soot on natural cirrus clouds is very small ($-6.9 \pm 29.8$ mW m$^{-2}$) and statistically insignificant at the 95% confidence level. For comparison, the total aviation ERF estimated by Lee et al. (2021) amounts to 100.9 mW m$^{-2}$ (with an uncertainty range between 55 and 145 mW m$^{-2}$). Further sensitivity simulations to analyse the role of other INPs (such as soot from other sources, mineral dust and ammonium sulphate) show that these largely control the microphysical and radiative impact of the heterogeneous freezing process on cirrus clouds, such that the impact of aviation soot remains negligible when the properties of these other INPs are varied, even under relatively bold assumptions weakening the effectiveness of these INPs in favour of aviation soot.

We conclude that the ERF impact of aviation soot on natural cirrus clouds is likely very small, thus confirming most previous studies, but for the first time with the support of laboratory measurements specifically targeting aviation soot and its ice nucleation ability. Future studies should therefore focus on the aviation-aerosol-interactions with low-level clouds in the liquid phase, where the impact of aviation-induced particles on cloud droplet number concentration could be relevant, resulting in a potentially significant climate effect (Gettelman and Chen, 2013; Righi et al., 2013; Kapadia et al., 2016; Righi et al., 2023).

*Code and data availability.* MESSy is continuously developed and applied by a consortium of institutions. MESSy and the source code are licensed to all affiliates of institutions which are members of the MESSy Consortium. Institutions can become members of the MESSy Consortium by signing the MESSy Memorandum of Understanding. More information can be found on the MESSy Consortium website (http://www.messy-interface.org, last access: 2 June 2025). The model configuration discussed in this paper is based on EMAC version 2.55. The output of the model simulations discussed in this paper is available at https://doi.org/10.5281/zenodo.15495975 (Righi, 2025).

*Author contributions.* MR designed and performed the simulations, analysed the model results, and wrote the paper. BT and ZAK provided the measurement data, contributed to the interpretation of the results and to the writing. CGB and JH contributed to the model simulations, to the interpretation of the results and to the writing. The study was conceived by all authors.

*Competing interests.* The authors declare no competing interests.

*Acknowledgements.* This study was supported by the European Commission via their Horizon 2020 Research and Innovation Programme (ACACIA project, grant no. 875036). We are grateful to Elena De La Torre Castro (DLR) for her helpful comments on an earlier version of the manuscript. Mattia Righi thanks the Writing Club of the DLR-PA-ESM department, for providing a pleasant working environment where this paper could be finalised, and Laura Stecher for organising it. The model simulations and data analysis for this work used the resources of the Deutsches Klimarechenzentrum (DKRZ) granted by its Scientific Steering Committee (WLA) under project ID bb1393 and the emission datasets provided by CMIP6 via the DKRZ data pool.

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
