# Peer review of "Aviation soot is unlikely to impact natural cirrus clouds"

_EGUsphere, 2025_

## Author Comment (AC1)

We would like to express our gratitude to the referees for their constructive comments, who helped us to revise and improve the manuscript. A detailed point-by-point reply can be found below: referees' comments in *italic blue*, replies in plain text, text passages quoted from the paper in red.

**Referee #1**

My general impression is that this paper nicely describes the work done, but that the model may be insufficient to actually give an accurate assessment or approximation of how aircraft emissions of soot actually affect cirrus clouds. This criticism is based mainly on atmospheric observations of two aspects: 1) cirrus clouds in regions heavily travelled by aircraft compared to regions with less aircraft travel (as noted in work by Urbanek, Gross and their colleagues), and 2) observations of aerosols in the atmosphere. Unfortunately, these latter observations are not specifically comparing regions heavily travelled by aircraft to regions with less aircraft travel, but they do show that aerosols that act as INP in the free atmosphere is made up of soot mixed with other species and is primarily coated by organics (Lata et al., 2021, China et al., JGR, 2017). This leads me to think that the observations of ice number concentrations by Urbanek and colleagues may be the result of mixing with organics. I read the papers described in the current pre-print, and the model apparently does not include coagulation or condensation of organics in the free atmosphere on the aircraft soot. Thus, the response in the model to emissions of aircraft soot may not be the response expected in the atmosphere. Admittedly, if the INP activity seen in the atmospheric observations is due to the coating by organics, then there may be a question of whether to assign the INP activity to the organics or to the primary soot emissions, but in any case, the soot from aircraft emissions may act as an INP after coating by organics (even though idealized laboratory experiments do not show this).

We are grateful to the referee for raising the point about the Urbanek et al. and Groß et al. studies. This has been addressed by including a dedicated discussion section (Sect. 4) to discuss the discrepancy with our result, also to address a similar concern by Referee #3. The new section also covers the referee's second point on the representativeness of the laboratory measurements for atmospheric conditions and about the role of organics. Further details can be found in our replies to the referee's specific comments below.

We also note that the model does in fact explicitly include coagulation and condensation of organics in the free troposphere on the aircraft soot and that the role of coating on the ice nucleating abilities of soot is represented in the cirrus parametrization. We have rephrased the corresponding statement in the Methods Section to further clarify this important point: "Since MADE3 explicitly simulates three different aerosol mixing states (soluble, insoluble and mixed; Kaiser et al., 2019), we introduced an important update in this study, by only allowing aviation soot in the insoluble modes of MADE3 to act as INP, consistent with the measurement results of coating-free soot described above indicating that only sulphur-free (uncoated) soot particles nucleate ice below the homogeneous freezing threshold, while in Righi et al. (2021) both insoluble and mixed aviation soot were allowed as INPs"

The paper does not provide the spatial distribution of the cirrus ERF induced by aircraft soot at all—all results are presented as global averages. Given that both flight routes and cirrus clouds are highly unevenly distributed globally, even if the study concludes no significant effect on a global average, could there still be regions with dense flight routes where the ERF becomes significant?

As discussed in our previous study (Righi et al., 2021), resolving the cirrus ERF at the geographical level is challenging due to statistical significance. That was the reason for showing only zonal means in that study. In the present study, given the non-significant impact of aviation soot, this is even more challenging and we had to limit the discussion to global results only (and even these are not significant). Nevertheless, to answer this question (also raised by Referee #2 below), we have included maps and zonal profiles of ERF in the supplement: these show a very noisy pattern without any coherent structure that could point to a significant effect at the local level. The concluding statement at the end of the Results section has also been extended accordingly: "It is therefore very challenging to draw a coherent picture, as the effect of aviation soot on natural cirrus clouds is very small compared to the internal model variability. This is also the case when repeating the analysis with a geographically-resolved simulation output: as shown in the maps and zonal profiles of the ERF effect provided in the Supplement, these are characterized by an extremely noisy pattern and no coherent picture emerges that could point to a significant ERF at the local level (e.g., over the North Atlantic or in other highly travelled regions of the world). The model results do not rule out the possibility of a localized effect, but they do indicate that it is likely to be small compared to natural climate variability, and therefore, the overall climate impact is expected to be minimal."

I am curious whether, based on the methodology and results of this study, the cirrus ERF induced by non-aircraft soot emissions would also be evaluated as insignificant. Similarly, would the cirrus ERF attributed to dust alone as an INP be significant, given that the model settings assign dust far superior ice nucleation capabilities compared to other INPs?

The ice nucleation abilities of non-aviation soot are also quite limited and, based on the variation study by Righi et al. (2021), we can presume that the corresponding ERF would be low, too. Note, however, that non-aviation soot reaching the UTLS is aged and that coating might further inhibit its ice nucleation abilities (this is also stated in the manuscript "As these are ground-based sources, it is reasonable to assume that the soot particles emitted by these sources are relatively aged when reaching the upper troposphere (Bond et al., 2013), i.e. the main region of interest for investigating aviation effects."). As argued by the referee, mineral dust is likely to have a significant effect due to its very good ice nucleation abilities. The combined ERF from dust and soot INPs was quantified in a previous study by our group (Beer et al., 2024).

In the ERF attributed to aircraft soot in this study, how much originates from the suppression of homogeneous ice nucleation, and how much from the enhancement of heterogeneous ice nucleation? Could the authors provide these results and separately verify whether both components are statistically insignificant?

This is already discussed in the manuscript and shown in Fig. 3e, depicting the impact of aviation soot on the homogeneous freezing fraction, which is not statistically significant for any of the experiments. The corresponding text passage is the following: "The impact on the homogeneous freezing fraction is negligible (Fig. 3e; see Righi et al., 2021, for the definition of this quantity), indicating that, on the global mean, aviation soot does not prevent the homogeneous formation of ice crystals, but just competes against other INPs for heterogeneous freezing. The very low statistical significance of this effect, however, suggests that with such low nucleation ability, aviation soot has little chance to compete against other more effective INPs for available supersaturated water vapour."

Line 58-59: statement is not necessarily true in the atmosphere.

The results from Testa et al (2024a, b) are in fact the only measurements of ice nucleation on real aviation soot emitted from aircraft engines. No other such measurements have been reported in the literature. Previous studies including our own were all done with proxies of soot using propane as a fuel, or kerosene burners with different air to fuel ratios, but not soot emitted from commercial in-use aircraft engine. We are not aware of any other measurements on ice nucleation specific to aviation soot. However, we modified the statement to now read "To the best of our knowledge, these constitute the most representative measurements on the ice nucleation ability of aviation engine emitted soot" We have removed the word in situ because that may imply we were chasing aircraft to sample the soot emissions in situ.

In our experiments engines that are currently in use were brought in for emissions testing and were run at idle, taxing, low, med and high thrust conditions when we sampled soot particles to test their ice nucleation ability. We sampled soot particles with organics and sulphur on them as well as by removing the organics and sulphur. The former being more atmospherically representative.

**Line 77-78: Statement is too strong and not necessarily true for aircraft soot in the atmosphere.**

As emphasized in Testa et al. (2024a, b), the number and size of cavities within the soot aggregates are the primary controlling factors of soot ice nucleation via PCF. This is primarily controlled by the primary particle morphology and the aggregates size. Gas phase chemistry and particle oxidation is thought to considerably slow down as the exhaust exits the combustor chamber, due to lower temperatures in the exhaust nozzle and downstream of the engine (Wong et al., 2008). As such, primary particle properties are fixed in the combustor (Dakhel et al., 2007). Such drop in temperature is also present in the sampling system (thousand degrees to 160°C), thus the primary particle overlap, size, crystallinity and oxidation are thought to be comparable to in situ emitted aviation soot particles. Besides, we note that (relatively high) temperature and pressure in the combustor (~2000 K and ~106 Pa, respectively; Dakhel et al., 2007; Starik, 2007) are largely driven by the engine design and thrust, hence uncorrelated to the ambient conditions.

Regarding the atmospheric relevance of the soot aggregate sizes in the measurement studies (mode diameter of 80-450 nm), airborne in situ measurement of soot sizes in young aircraft plumes are scarce: Petzold et al. (1999); Petzold et al. (1998) and Twohy and Gandrud (1998) observed interstitial and contrail residual soot aggregates of 0.15-1 µm. Those large aggregates could result from the coagulation of soot aggregates trapped in the wingtip vortices (Miake-Lye R. C. et al., 1993) due to the higher soot emission index for older engine models (Lee et al., 2010; Masiol & Harrison, 2014). Coagulation of contrailice crystals and merging of embedded soot aggregates upon sublimation of the ice crystals could also lead to larger soot aggregates. Yet, for current aircraft engines with lower emission indices, coagulation of the soot aggregate is reduced or inhibited, as shown by Moore et al. (2017), who did not observe soot coagulation, and whose measurements are similar to what would be measured at ground in engine maintenance and testing facilities (e.g., Durdina et al., 2021). Thus, under ambient conditions, coagulation is likely not expected. Hence atmospheric aircraft soot has very likely smaller mode diameters than soot aggregates sampled in the measurement study, which would further limit the potential to act as INP at RH below homogeneous freezing (size dependency of soot particles has been shown in numerous studies before, e.g., K. Gao et al., 2022; Mahrt et al., 2018; Zhang et al., 2020).

We prefer to keep the statement as is, because the particles we sample here are much more representative than previously studied particles from propane/acetylene/kerosene burners and

commercially available soot and black carbon sources hat claimed to be representative of aircraft soot (DeMott, 1990; DeMott et al., 1999; Koehler et al., 2009; Nichman et al., 2019; Zhang et al., 2020), but we have added a paragraph in the new Discussion Sect. 4 to summarize the above arguments.

Line 82-84: Here, I believe, this really depends on the organics that were tested and whether they agree with the mixtures expected in the atmosphere.

The dependence on the organic composition maybe true if pure organics coated the soot particles without inorganics. In the cirrus regime, ice nucleation activity of organics is mostly attributed to their phase state being glassy and thus if the temperature of nucleation is below the glassy transition temperature, the organic coatings or particles may nucleate ice heterogeneously, assuming the organics turn glassy before water vapor can diffuse into the organic coating as has been shown by model organic substances and mixtures in laboratory experiments (Murray et al., 2010; Wilson et al., 2012). This has also been shown for mixed organic particles sampled from the field (boundary layer polluted conditions) and tested in the laboratory (Wang et al., 2012). However, for aircraft soot in the atmosphere, the presence of sulphate mixed with the organics results in water uptake thus resulting in homogeneous freezing. If the organics were largely dominant with the absence of sulphate or nitrate in the troposphere, it would be believable that organic coatings turn glassy at low RHw and low T being able to act as glassy heterogeneous INP. But the presence of sulphate promotes water uptake and homogeneous freezing. This has been clearly shown in Testa et al. (2024a), where aviation soot particles, whose sulphate and organics were not stripped, froze homogeneously.

Line 107-108: This is a key assumption that may not be true in the atmosphere. Also, would there not be coating of sulfate and organics that takes place within the aircraft plume?

We agree that soot can interact with aircraft condensable vapors (mainly sulphur and organic vapor) and with nucleation mode particles (H2SO4 and organic oil droplets) in the young aircraft plume. This was suggested by multiple studies, despite some conducted at ground level, where ambient temperature and dilution dynamic might differ from high altitude plume (Kärcher et al., 2007; Onasch et al., 2009; Ungeheuer et al., 2022; Wong et al., 2014; Yu et al., 2014). Differences in temperature and dynamics between the experimental setup used for the measurement and real atmospheric conditions are as follows. In the aerosol reservoir, the temperature and pressure are higher, no nucleation mode particle (H2SO4 and organic droplets e.g., oil droplets) is allowed to form and no dilution with ambient air takes place. These factors/processes would likely affect the soot mixing state, presumably decreasing the coating over the soot particles (Kärcher et al., 2007; Onasch et al., 2009; Peck et al., 2014; Timko et al., 2013; Wong et al., 2014; Wong et al., 2008) for soot sampled in the ground-based study. Differences in equilibrium temperatures and dilution would nonetheless impact the partitioning of volatile unburned hydrocarbons and sulphur compounds, i.e., the soot mixing state, and hence the soot ice nucleation abilities. The higher temperatures of the ground-based setup (160°C and then room temperatures) compared to the temperature at which the exhaust would be exposed in the atmosphere (- 60°C) would reduce the condensation of volatiles onto the soot due to their higher equilibrium saturation pressures. Higher temperature together with the drying of the exhaust before sampling into the aerosol tank would also prevent the formation of nucleation mode particles. On the other hand, reduced particle surface area (no nucleation mode particles) would favor condensation of volatiles onto the soot particles. Besides, the gases only get slowly diluted by synthetic air in the aerosol reservoir, compared to the strong in situ dilution of the exhaust gases that occur within the first seconds in the atmosphere (Kärcher et al., 2007). Nonetheless, modelling studies (Kärcher, 1998; Kärcher et al., 2007; Yu et al.,

1999) showed that nucleation mode particles and aviation soot are thought to interact in the young aircraft plume downstream of the engine, increasing the soot coating. This process did not take place in the ground-based measurement due to the absence of nucleation mode particles.

We have summarized this in the Discussion section as follows: "We expect in situ particles to be coated with sulphuric acid and organics. To which extent the coating of the aviation soot sample in the measurement study is different from in situ aviation soot cannot be quantified, but any condensation of organics or sulphate in the atmosphere would first condense into soot pores which inhibits ice nucleation (Gao et al., 2022; Gao & Kanji, 2022). Thus, the ice nucleation abilities of coating-free and coated soot were quantified in the measurement study to bound the possible effect of different aviation soot mixing states. We note that a larger or smaller soot coating compared to the investigated coated soot sample would favour homogeneous nucleation, or be bound by a coating-free soot sample, respectively. Finally, as mentioned above, ice nucleation is controlled by the limited availability and morphology of inter-aggregate cavities, primarily governed by the primary particle morphology, which is not affected by the experimental setup and sampling method but rather fixed in the engine combustor."

Line 125-133: What are the length of the simulations? Longer simulations could reduce the uncertainty. Also, the simulations appear to me to be nudged towards observed (or ECMWF) temperatures and velocities (based on the magnitude of uncertainty reported here, in comparison to the Righi, et al 2020 paper). You need to state this, if true, or say these are free-standing model predictions.

Thanks for noting the missing information. It has been added at the end of Sect. 2.2: "All simulations cover a period of 15 years (2001–2015) preceded by a spin-up year (2000) not considered for the analysis [...] To reduce the impact of the internal model variability when comparing two simulations, the model meteorology (temperature, winds, and logarithm of the surface pressure) is nudged towards the ERA-Interim reanalysis data (Dee et al., 2011) of the European Centre for Medium-Range Weather Forecasts (ECMWF) for the simulation period."

Line 150: I don't believe this statement is true, since the ice nucleation properties, as measured in the laboratory, may not represent their properties in the atmosphere.

The measurements used in the current studies are the best estimates of the ice nucleation activates of real aviation soot and are the first representation of constraining these estimates compared to anything else in the literature. Furthermore, we propose that our studies in fact provide an upper estimate of ice nucleation abilities and in the absence of coagulation. In the newer lower emission index engines, the size of the soot aggregates peaks at around 40 nm, which in all types of soot explored in the literature is way too small to be ice nucleation active. Even for the most efficiency soot samples and proxies, anything below 200 nm only nucleated ice homogeneously simply due to the absence of enough porous cavities (K. Gao et al., 2022a; Mahrt et al., 2018).

Line 155-156: aged soot can act as good INP in the atmosphere, as shown by Lata et al.

We can refer to the paragraph in Testa et al. (2024b): "In addition to contrail processing, several soot aging processes can occur in the atmosphere, such as interaction with background aerosols and volatile compounds or oxidation of aerosols by  $O_3$  and OH radicals (Bond et al., 2013). Interception of soluble aerosol onto the soot surface would increase the amount of soot coating preventing PCF. Exposure of aviation soot to  $O_3$  or OH at flight altitude can cause

condensed organics to desorb due to the breaking of covalent bonds with the elemental soot fraction. The oxidized organics could re condense onto the soot due to their lowered volatility, e.g., short alkanes and aldehyde that are soluble in acidic solution (e.g.,  $H_2SO_4$  solution; Yu et al., 1999), and could lead to a freezing point depression if condensed in pores or prevent water uptake by blocking the pores if they are hydrophobic, inhibiting PCF (KF Gao and Kanji, 2024; K. Gao et al., 2022b). Nevertheless, we do not rule out the possibility of the oxidation of soot surface organics also leading to glassy coatings and promoting deposition ice nucleation of the particles (Tian et al., 2022)."

Furthermore, the comparison to Lata et al. (2021) is misguided for several reasons. These were boundary layer aerosols that are heavily internally mixed because of long range transport, mixed in with marine emissions that could result in internal mixing with biogenic or biological components as well as heavily organically polluted airmasses as shown in the composition spectra and size resolved spectra of the paper in Figure 2 (a-c). The size of the particles here should be kept in mind as well. Ice nucleation is a size dependent process and probabilistic as well (stochastic). The large particle sizes used in this study indeed make it probably that elemental carbon could nucleate ice. But in aircraft corridors where sub-100 nm particles are emitted, there is a strong size component suppressing heterogenous ice nucleation that is well documented in the literature as cited in the numerous references above.

Line 213: this experimentally informed soot-cirrus impact does not account for atmospheric observations.

We agree with the reviewer that this assessment does not account for atmospheric observations but just experiments on real aviation soot emissions (not proxies) as we expressed in the paper in the initial manuscript (preprint version) on line 211-213.

**Referee #2**

We thank the referee for pointing out some missing information about the evaluation of the statistical significance of our results. This has now been added and the other referee's comments have been considered.

I do think some important additional work should be done before publication: some map figures illustrating regional effects would help to understand what is noise and what is signal.

To address this point (also raised by Referee #1 above), we have included maps and zonal profiles of ERF in the supplement: these show a very noisy pattern without any coherent structure that could point to a significant effect at the local level. The concluding statement at the end of the Results section has been extended accordingly to include this information.

Page 5, L112: is Scrit fixed in the model for each aerosol type? Shouldn't Scrit vary for each aerosol type as a function of temperature?

For dust INP a fitting function accounting for the temperature dependence of  $S_{crit}$  (Möhler et al., 2006) is included in the model, but for other INPs the cirrus parametrization considers only a single threshold value. In a pilot study performed for this work, we attempted to mimic a temperature spectrum for  $S_{crit}$  of aviation soot by introducing 3 activation points in the cirrus parametrization at different active fraction (0.001, 0.01, 0.1) and found that this would not significantly impact the conclusions.

Page 6, L125: have you stated how long the simulations are? Free running or nudged. I think you need a bit more detail.

This information was missing and has been added in the methods section. Thanks for spotting this.

Page 6, L129: See comment above. How do you pick a small signal for ΔERF out of the noise?

We apply a paired-sample t test, as explained in the Methods section: "To validate the statistical significance of the results, a paired-sample t test is applied. The results are considered significant if the null hypothesis that the paired simulations are identical can be rejected at a confidence level larger than 95%. The same methodology is applied when other model variables are evaluated, such as the aviation-soot-induced changes in ice crystal number concentration (ICNC), total water (water vapour plus ice water) and cloud frequency."

Page 6, L140: To understand whether this is noise or not, it would be useful to show maps of  $\Delta$ ERF (SW and LW) for one or more of the experiments: is there an expected pattern to  $\Delta$ ERF that looks like aviation, or is it really noise?

As mentioned above, the maps (now included in the Supplement) show indeed that this is only noise and no aviation pattern can be identified. We tried this both with lat-lon maps and zonal profiles and no coherent pattern is visible in either case. This is also the reason why we decided to show only aggregated results. See also the Supplement of our previous study, Righi et al., 2021, where we were confronted with a similar issue even when assuming very good ice nucleating abilities for aviation soot: even in that case statistical significance was only found at the globally aggregated level.

Page 7, L164: how does no change in ICNC in the NOBGSOOT lead to the same change in SW? You explain why there is no change to LW, but then why is there still a SW change?

We should be careful not to overinterpret these results, as in both cases the signal is not statistically significant and is characterized by a very large interannual variability. Hence, it is not possible to draw any conclusion by the comparison. We also noted this at the end of Sect. 3: "In summary, the quantification of the aviation-soot cirrus effect with the support of novel laboratory measurements on the ice nucleating properties of aviation soot result in a non-statistically significant ERF effect for all investigated cases. The interpretation of the model results in relation to key cloud and radiation variables is substantially hampered by the very low statistical significance of almost all discussed quantities and in all simulations. It is therefore very challenging to draw a coherent picture, as the effect of aviation soot on natural cirrus clouds is very small compared to the internal model variability."

Page 10, L208: as noted, it would be interesting to look at some maps of these quantities, at least for the baseline case with background aerosols. Are the changes significant in flight regions?

See above reply, maps are now included in the Supplement and briefly commented at the end of the Results section.

**Referee #3**

We are grateful to the referee for their detailed comment on the PLDR studies, an issue also raised by Referee #1. We are aware of these studies, although we only cited Urbanek et al. in the introduction. This has now been extended to include Li and Groß (2021), Li and Groß (2022) and Groß et al. (2023).

We agree with the referee that these studies are not consistent with our conclusions, but we would not call them "evidences", as they do not demonstrate a causal relationship between increased depolarization ratios and aviation soot INPs. The laboratory measurements by Testa et al. used in our study, on the contrary, do provide a direct measurement.

For instance, Urbanek et al. concluded: "These results could be explained by an indirect aerosol effect where heterogeneous freezing is caused by aviation exhaust particles", while Li and Groß (2022) wrote: "changes in the properties and occurrence of cirrus clouds [...] which are presumed to be caused by the corresponding reduction in civil aviation, and again Groß et al. (2023): "We suspect that a suppression of homogeneous ice formation by the heterogeneously freezing soot aerosol particles included in the areas affected by air traffic is the cause of the reduced ice crystal concentrations."

So, at present, these studies suggest a hypothesis rather than providing evidence. Nevertheless, we acknowledge that this is a strong hypothesis that cannot be ignored in the context of our study and we have added a dedicated section (Section 4: Discussion) to elaborate on this conundrum and discuss possible explanation for the discrepancies. The references provided by the referee were very helpful and have been considered. Thank you again for the insights.

We also note that our study does not exclude that the above effect exists, but we cannot distinguish it from the internal model variability, which results in a statistically not significant climate effect.

In that sense, we also realized that the title of the manuscript could have been misleading and we have changed it to: "Aviation soot interactions with natural cirrus clouds are unlikely to have a significant impact on global climate"

Lines 24-25: If there is a reference for this statement, please provide one. Otherwise, the statement is obvious enough to overlook this need.

Reference to DeMott et al. (2003) and Kärcher and Lohmann (2003) added.

Lines 29 – 33: It may be of interest that orographic gravity waves (OGWs) are not represented in the NCAR climate models (Lyu et al., 2023, JGR). OGWs appear to contribute substantially to cirrus coverage, promoting hom due to their higher updrafts (Barahona et al., 2017, Nature; Gryspeerdt et al., 2018, ACP; Mitchell et al., 2018, ACP).

Thanks for this suggestion, it has been added.

Line 142: Fig. 3f => Fig. 3e?

Yes, fixed.

Figure 2 caption: (ERF => ERF?

Bracket removed.

**Referee #4**

We are grateful to the referee for their suggestions, which were in line with the comments by the other referees. These have been addressed below and greatly helped to improve the manuscript.

Section 2.1: The description of contrail processing simulations for sampled aviation soot lacks details on the experimental conditions (e.g., temperature, pressure, duration) and how these parameters mimic real atmospheric contrail formation.

We agree that our setup does not mimic the dynamical processes and rates relevant for contrail formation, e.g., exhaust temperature dropping rapidly (< 1 s) from thousands of degrees to  $-60\,^{\circ}\text{C}$  and pressure from  $\sim 10^6$  Pa to below  $10^5$  Pa (Kärcher, 2018). Instead, the time scales are longer that what it would be in the atmosphere, the pressure drops from  $\sim 10^6$  Pa in the engine, to  $\sim 10^5$  Pa (atmospheric pressure in our measurement set up). The temperature also drops in three steps, from the engine temperature to the heated line  $160\,^{\circ}\text{C}$  and then to room temperature followed by a third drop from room temperature to the cloud chamber temperature. So, we do have similar temperature drops as would occur in the atmosphere, but they occur over a slightly longer time scale.

We have now added the above description into the methods to clarify the differences between our sampling method from the atmospheric process. In addition, we address this in the new discussion section added to the revised paper as well.

Section 2.2: The model simulations (Table 1) involve scaling factors for mineral dust INPs (e.g., NOBGSOOT+DUST5/10) to correct for a positive bias in upper tropospheric dust concentrations. However, the basis for choosing factors of 5 and 10 is not explicitly justified. Please provide quantitative evidence (e.g., comparison with observational data) supporting these specific scaling values.

The choice of factors 5 and 10 is based on a quantitative comparison with observational data in Beer et al. (2024). This is mentioned in the paper: "In the NOBGSOOT+DUST5 and NOBGSOOT+DUST10 the contribution of mineral dust INPs to the immersion and deposition mode is reduced, by scaling fact down by a factor 5 and 10, respectively. These two simulations aim to account for a potential positive bias of EMAC in the representation of mineral dust concentration in the upper troposphere (see Fig. S1 in Beer et al., 2024)."

Section 3: The statistical significance assessment using a paired-sample t-test is mentioned, but details on the sample size (e.g., number of model years, spatial grid points) and how internal model variability was quantified are lacking. This is critical for evaluating the robustness of the "non-significant" conclusion, please expand on the statistical methodology.

Thanks, this point was also raised by the other referees and has been addressed in the Methods section: "All simulations cover a period of 15 years (2001–2015) preceded by a spin-up year (2000) not considered for the analysis. [...] To reduce the impact of the internal model variability when comparing two simulations, the model meteorology (temperature, winds, and logarithm of the surface pressure) is nudged towards the ERA-Interim reanalysis data (Dee et al., 2011) of the European Centre for Medium-Range Weather Forecasts (ECMWF) for the simulation period."

Fig. 4: The relative share of ice crystals from aviation soot remains at 0.04% across REF and NOBGSOOT simulations, but the underlying mechanism for this stability is not explained. Given the removal of background soot in NOBGSOOT, why does aviation soot not increase its contribution? A more detailed microphysical explanation is needed.

As noted above to an analogous question by Referee #2, we should be careful not to overinterpret these results, as in both cases the signal is not statistically significant and is characterized by a very large interannual variability. Hence, it is not possible to draw any conclusion by the comparison. We also noted this at the end of Sect. 3.

Section 3.3: The discussion of ammonium sulfate as an INP (NOBGSOOT+AMSU) notes that its effect is transient due to efflorescence/deliquescence, but there is no quantitative analysis of its temporal variability. How does this transience affect the model's ability to capture its competition with aviation soot?

In the paper, we discuss only mean values of periods of crystalline and liquid ammonium sulphate, but the model can explicitly capture this temporal variability and the resulting competition with aviation soot. The different regimes are considered but not discussed as we focused on long term means, which are the relevant quantities for analysing climate effects (and also because of statistical significance).

The comparison with prior studies (e.g., NCAR CAM model results showing large ERF vs. ECHAM4/CAM5/CESM2 with non-significant effects) is cursory. The authors should explicitly discuss why their experimental constraints resolve this discrepancy.

Our experimental constrain helps to remove one of the possible reasons for discrepancy, but does not completely solve it. We note, however, that the discrepancy exists only in comparison with the NCAR CAM model, whereas our conclusions are consistent with all other previous modelling studies. This is the reason why we concluded that aviation soot is *unlikely* to impact global climate. Without knowing the details of the NCAR CAM model and the simulation setup used in the related studies, it is not possible to argue on other possible reason for discrepancy. A model intercomparison would only make sense if a consistent modelling protocol were defined and is anyway beyond the scope of this work.

The conclusion suggests future focus on aviation-aerosol interactions with low-level clouds, but the manuscript does not address the limitations of the current model in simulating such interactions. How might the model's cirrus parameterization differ from that of low-level clouds, and does this affect the transferability of the methodology?

The aviation-aerosol effect on low-level clouds is mainly related to sulphate aerosol (see e.g., Righi et al. 2013; Gettelman and Chen, 2013), whose ability to act as a CCN is much better constrained than the aviation-soot ice nucleating ability. The involved processes (cloud activation vs. heterogenous ice formation) are also completely different, hence these two effects (soot and sulphate) are essentially decoupled. Although the cloud scheme of EMAC adopted here (Righi et al., 2020) can consistently simulate cloud microphysics for all phases, a cirrus parametrization is not required to quantify the low-cloud effect.

Section 2.2: The model update restricting aviation soot INP activity to the insoluble mode (vs. Righi et al., 2021) is justified by laboratory data on coating-free soot. However, the fraction of ambient aviation soot that remains uncoated in the upper troposphere is not quantified. Please provide estimates of this fraction to support the model's assumption.

This is not an assumption: the model explicitly represents and distinguishes between coated and uncoated soot, via the aerosol microphysical processes (condensation and coagulation). We have rephrased the corresponding statement in the Methods section, to make clearer that the cirrus parametrization consistently considers only uncoated soot to act as an INP: "Since MADE3 explicitly simulates three different aerosol mixing states (soluble, insoluble and mixed;

Kaiser et al., 2019), we introduced an important update in this study, by only allowing aviation soot in the insoluble modes of MADE3 to act as INP, consistent with the measurement results of coating-free soot described above indicating that only sulphur-free (uncoated) soot particles nucleate ice below the homogeneous freezing threshold, while in Righi et al. (2021) both insoluble and mixed aviation soot were allowed as INPs"

**References**

Beer, C. G., Hendricks, J., and Righi, M.: Impacts of ice-nucleating particles on cirrus clouds and radiation derived from global model simulations with MADE3 in EMAC, Atmos. Chem. Phys., 24, 3217–3240, https://doi.org/10.5194/acp-24-3217-2024, 2024.

Dakhel, P. M., Lukachko, S. P., Waitz, I. A., Miake-Lye, R. C., and Brown, R. C.: Postcombustion Evolution of Soot Properties in an Aircraft Engine, J. Propul. Power, 23, 942–948, <a href="https://doi.org/10.2514/1.26738">https://doi.org/10.2514/1.26738</a>, 2007.

DeMott, P. J., An Exploratory-Study of Ice Nucleation by Soot Aerosols, Journal of Applied Meteorology, 29(10), 1072-1079. <a href="https://doi.org/10.1175/1520-0450(1990)029%3C1072:AESOIN%3E2.0.CO;2">https://doi.org/10.1175/1520-0450(1990)029%3C1072:AESOIN%3E2.0.CO;2</a>, 1990.

DeMott, P. J., Y. Chen, S. M. Kreidenweis, D. C. Rogers, and D. E. Sherman, Ice formation by black carbon particles, Geophys. Res. Lett., 26(16), 2429-2432. https://doi.org/10.1029/1999GL900580, 1999.

Durdina, L., Brem, B. T., Elser, M., Schönenberger, D., Siegerist, F., and Anet, J. G.: Reduction of Nonvolatile Particulate Matter Emissions of a Commercial Turbofan Engine at the Ground Level from the Use of a Sustainable Aviation Fuel Blend, Environ. Sci. Technol., 55, 14 576–14 585, <a href="https://doi.org/10.1021/acs.est.1c04744">https://doi.org/10.1021/acs.est.1c04744</a>, 2021.

Gao, K., and Z. Kanji, Influence of Lowering Soot-Water Contact Angle on Ice Nucleation of Ozone-Aged Soot, Geophys. Res. Lett., 51(7), doi:10.1029/2023GL106926. https://doi.org/10.1029/2023GL106926, 2024.

Gao, K., F. Friebel, C. W. Zhou, and Z. A. Kanji, Enhanced soot particle ice nucleation ability induced by aggregate compaction and densification, Atmos. Chem.Phys., 22(7), 4985-5016, <a href="https://doi.org/10.5194/acp-22-4985-2022">https://doi.org/10.5194/acp-22-4985-2022</a>, 2022.

Gao, K., C. W. Zhou, E. J. B. Meier, and Z. A. Kanji, Laboratory studies of ice nucleation onto bare and internally mixed soot-sulfuric acid particles, Atmos. Chem. Phys., 22(8), 5331-5364, https://doi.org/10.5194/acp-22-5331-2022, 2022b.

Kärcher, B., On the potential importance of sulfur-induced activation of soot particles in nascent jet aircraft exhaust plumes, Atmos. Res., 46, 3–4, 293-305, <a href="https://doi.org/10.1016/S0169-8095(97)00070-7">https://doi.org/10.1016/S0169-8095(97)00070-7</a>, 1998.

Kärcher, B., Möhler, O., DeMott, P. J., Pechtl, S., and Yu, F.: Insights into the role of soot aerosols in cirrus cloud formation, Atmos. Chem. Phys., 7, 4203–4227, https://doi.org/10.5194/acp-7-4203-2007, 2007.

Koehler, K. A., P. J. DeMott, S. M. Kreidenweis, O. B. Popovicheva, M. D. Petters, C. M. Carrico, E. D. Kireeva, T. D. Khokhlova, and N. K. Shonija, Cloud condensation nuclei and ice nucleation activity of hydrophobic and hydrophilic soot particles, Physical Chemistry Chemical Physics, 11(36), 7906-7920, doi:10.1039/b905334b. https://doi.org/10.1039/b905334B, 2009.

Lata, N. N., B. Zhang, S. Schum, L. Mazzoleni, R. Brimberry, M. A. Marcus, W. H. Cantrell, P. Fialho, C. Mazzoleni, and S. China, Aerosol Composition, Mixing State, and Phase State of Free Tropospheric Particles and Their Role in Ice Cloud Formation, ACS Earth and Space Chemistry, 5(12), 3499-3510, doi:10.1021/acsearthspacechem.1c00315. https://doi.org/10.1021/acsearthspacechem.1c00315, 2021.

Lee, D., Pitari, G., Grewe, V., Gierens, K., Penner, J., Petzold, A., Prather, M., Schumann, U., Bais, A., and Berntsen, T.: Transport impacts on atmosphere and climate: Aviation, Atmos. Environ., 44, 4678–4734, <a href="https://doi.org/10.1016/j.atmosenv.2009.06.005">https://doi.org/10.1016/j.atmosenv.2009.06.005</a>, 2010.

Mahrt, F., C. Marcolli, R. O. David, P. Gronquist, E. J. B. Meier, U. Lohmann, and Z. A. Kanji, Ice nucleation abilities of soot particles determined with the Horizontal Ice Nucleation Chamber, Atmos. Chem. Phys., 18(18), 13363-13392, doi:10.5194/acp-18-13363-2018. https://doi.org/10.5194/acp-18-13363-2018, 2018.

Masiol, M. and Harrison, R. M.: Aircraft engine exhaust emissions and other airport-related contributions to ambient air pollution: A review, Atmos. Environ., 95, 409–455, <a href="https://doi.org/10.1016/j.atmosenv.2014.05.070">https://doi.org/10.1016/j.atmosenv.2014.05.070</a>, 2014.

Miake-Lye, R. C., Martinez-Sanchez, M., Brown, R. C., and Kolb, C. E.: Plume and wake dynamics, mixing, and chemistry behind a high speed civil transport aircraft, J. Aircraft, 30, 467–479, <a href="https://doi.org/10.2514/3.46368">https://doi.org/10.2514/3.46368</a>, 1993.

Moore, R. H., Thornhill, K. L., Weinzierl, B., Sauer, D., D'Ascoli, E., Kim, J., Lichtenstern, M., Scheibe, M., Beaton, B., Beyersdorf, A. J., Barrick, J., Bulzan, D., Corr, C. A., Crosbie, E., Jurkat, T., Martin, R., Riddick, D., Shook, M., Slover, G., Voigt, C., White, R., Winstead, E., Yasky, R., Ziemba, L. D., Brown, A., Schlager, H., and Anderson, B. E.: Biofuel blending reduces particle emissions from aircraft engines at cruise conditions, Nature, 543, 411–415, <a href="https://doi.org/10.1038/nature21420">https://doi.org/10.1038/nature21420</a>, 2017

Murray, B. J., et al., Heterogeneous nucleation of ice particles on glassy aerosols under cirrus conditions, Nature Geosci., 3(4), 233-237, <a href="https://doi.org/10.1038/ngeo817">https://doi.org/10.1038/ngeo817</a>, 2010.

Nichman, L., M. Wolf, P. Davidovits, T. B. Onasch, Y. Zhang, D. R. Worsnop, J. Bhandari, C. Mazzoleni, and D. J. Cziczo, Laboratory study of the heterogeneous ice nucleation on black-carbon-containing aerosol, Atmos. Chem. Phys., 19(19), 12175-12194, doi:10.5194/acp-19-12175-2019. https://doi.org/10.5194/acp-19-12175-2019, 2019.

Onasch, T., Jayne, J.T, Herndon, S., Worsnop, D. R., Miake-Lye, R. C., Mortimer, I. P., and Anderson, B. E., Chemical Properties of Aircraft Engine Particulate Exhaust Emissions, J. Propul. Power, 25:5, 1121-1137, https://doi.org/10.2514/1.36371, 2009.

Peck, J., Yu, Z., Wong, H., Miake-Lye, R., Liscinsky, D., Jennings, A., and True, B. Experimental and Numerical Studies of Sulfate and Organic Condensation on Aircraft Engine Soot. Proceedings of the ASME Turbo Expo 2014: Turbine Technical Conference and Exposition. Volume 4A: Combustion, Fuels and Emissions. Düsseldorf, Germany. June 16–20, 2014. V04AT04A015. ASME. <a href="https://doi.org/10.1115/GT2014-25227">https://doi.org/10.1115/GT2014-25227</a>, 2014.

Petzold, A., Ström, J., Ohlsson, S., and Schröder, F.: Elemental composition and morphology of ice-crystal residual particles in cirrus clouds and contrails, Atmos. Res., 49, 21–34, <a href="https://doi.org/10.1016/S0169-8095(97)00083-5">https://doi.org/10.1016/S0169-8095(97)00083-5</a>, 1998.

Petzold, A., Döpelheuer, A., Brock, C. A., and Schröder, F.: In situ observations and model calculations of black carbon emission by aircraft at cruise altitude, J. Geophys. Res. Atmos., 104, 22 171–22 181, <a href="https://doi.org/10.1029/1999jd900460">https://doi.org/10.1029/1999jd900460</a>, 1999.

Righi, M., Hendricks, J., and Beer, C. G.: Exploring the uncertainties in the aviation soot–cirrus effect, Atmos. Chem. Phys., 21, 17267–17289, <a href="https://doi.org/10.5194/acp-21-17267-2021">https://doi.org/10.5194/acp-21-17267-2021</a>, 2021.

Starik, A. M.: Gaseous and particulate emissions with jet engine exhaust and atmospheric pollution, in: Advances on propulsion technology for high-speed aircraft, edited by Paniagua, G. and Steelant, J., vol. II of RTO-AVT-VKI Lecture Series, pp. 1–19, von Karman Institute for Fluid Dynamics, Belgium, <a href="https://apps.dtic.mil/sti/tr/pdf/ADA473844.pdf">https://apps.dtic.mil/sti/tr/pdf/ADA473844.pdf</a>, 2007.

Testa, B., Durdina, L., Alpert, P. A., Mahrt, F., Dreimol, C. H., Edebeli, J., Spirig, C., Decker, Z. C. J., Anet, J., and Kanji, Z. A.: Soot aerosols from commercial aviation engines are poor ice-nucleating particles at cirrus cloud temperatures, Atmos. Chem. Phys., 24, 4537–4567, <a href="https://doi.org/10.5194/acp-24-4537-2024">https://doi.org/10.5194/acp-24-4537-2024</a>, 2024a.

Testa, B., Durdina, L., Edebeli, J., Spirig, C., and Kanji, Z. A.: Simulated contrail-processed aviation soot aerosols are poor ice-nucleating particles at cirrus temperatures, Atmos. Chem. Phys., 24, 10409–10424, <a href="https://doi.org/10.5194/acp-24-10409-2024">https://doi.org/10.5194/acp-24-10409-2024</a>, 2024b.

- Tian, P., Liu, D., Bi, K., Huang, M., Wu, Y., Hu, K., Li, R., He, H., Ding, D., Hu, Y., Liu, Q., Zhao, D., Qiu, Y., Kong, S., and Xue, H.: Evidence for Anthropogenic Organic Aerosols Contributing to Ice Nucleation, Geophys. Res. Lett., 49, e2022GL099990, <a href="https://doi.org/10.1029/2022GL099990">https://doi.org/10.1029/2022GL099990</a>, 2022.
- Timko, M. T., Fortner, E., Franklin, J., Yu, Z., Wong, H.-W., Onasch, T. B., Miake-Lye, R. C., and Herndon, S. C. Atmospheric Measurements of the Physical Evolution of Aircraft Exhaust Plumes. Environ. Sci. Technol., 47:3513–3520, https://doi.org/10.1021/es304349c, 2013.
- Twohy, C. H. and Gandrud, B. W.: Electron microscope analysis of residual particles from aircraft contrails, Geophys. Res. Lett., 25, 1359–1362, https://doi.org/10.1029/97GL03162, 1998.
- Ungeheuer, F., Caudillo, L., Ditas, F. et al. Nucleation of jet engine oil vapours is a large source of aviation-related ultrafine particles. Commun. Earth Environ. 3, 319. <a href="https://doi.org/10.1038/s43247-022-00653-w">https://doi.org/10.1038/s43247-022-00653-w</a>, 2022.
- Wang, B. B., A. Laskin, T. Roedel, M. K. Gilles, R. C. Moffet, A. V. Tivanski, and D. A. Knopf, Heterogeneous ice nucleation and water uptake by field-collected atmospheric particles below 273 K, J. Geophys. Res.-Atmos., 117, D00v19, https://doi.org/10.1029/2012JD017446, 2012.
- Wilson, T. W., et al., Glassy aerosols with a range of compositions nucleate ice heterogeneously at cirrus temperatures, Atmos. Chem. Phys., 12(18), 8611-8632, <a href="https://doi.org/10.5194/acp-12-8611-2012">https://doi.org/10.5194/acp-12-8611-2012</a>, 2012.
- Wong, H.-W., Yelvington, P. E., Timko, M. T., Onasch, T. B., Miake-Lye, R. C., Zhang, J., and Waitz, I. A.: Microphysical Modeling of Ground-Level Aircraft-Emitted Aerosol Formation: Roles of Sulfur-Containing Species, J. Propul. Power, 24, 590-602, <a href="https://doi.org/10.2514/1.32293">https://doi.org/10.2514/1.32293</a>, 2008.
- Wong, H. W., Jun, M., Peck, J., Waitz, I. A., & Miake-Lye, R. C. Detailed Microphysical Modeling of the Formation of Organic and Sulfuric Acid Coatings on Aircraft Emitted Soot Particles in the Near Field. Aerosol Sci. Technol., 48(9), 981–995, https://doi.org/10.1080/02786826.2014.953243, 2014.
- Yu, F., R. P. Turco, and B. Kärcher, The possible role of organics in the formation and evolution of ultrafine aircraft particles, J. Geophys. Res., 104(D4), 4079–4087, <a href="https://doi.org/10.1029/1998JD200062">https://doi.org/10.1029/1998JD200062</a>, 1999.
- Yu, Z., Liscinsky, D. S., True, B., Peck, J., Jennings, A. C., Wong, H.-W., et al. Uptake Coefficient of Some Volatile Organics Compounds by Soot and Their Application in Understanding Particulate Matter Evolution in Aircraft Engine Exhaust Plumes. ASME J. Eng. Gas Turbines Power, 136: 121501, https://doi.org/10.1115/1.4027707, 2014.
- Zhang, C., Y. Zhang, M. J. Wolf, L. Nichman, C. Shen, T. B. Onasch, L. Chen, and D. J. Cziczo, The effects of morphology, mobility size, and secondary organic aerosol (SOA) material coating on the ice nucleation activity of black carbon in the cirrus regime, Atmos. Chem. Phys., 20(22), 13957-13984, <a href="https://doi.org/10.5194/acp-20-13957-2020">https://doi.org/10.5194/acp-20-13957-2020</a>, 2020.